# Adversarial Representation Engineering: A General Model Editing Framework for Large Language Models

Yihao Zhang[1,2]     Zeming Wei[1]     Jun Sun[2*]     Meng Sun[1*]

[1]Peking University     [2]Singapore Management University

## Abstract

Since the rapid development of Large Language Models (LLMs) has achieved remarkable success, understanding and rectifying their internal complex mechanisms has become an urgent issue. Recent research has attempted to interpret their behaviors through the lens of inner representation. However, developing practical and efficient methods for applying these representations for general and flexible model editing remains challenging. In this work, we explore how to leverage insights from representation engineering to guide the editing of LLMs by deploying a representation discriminator as an editing oracle. We first identify the importance of a robust and reliable discriminator during editing, then propose an **A**dversarial **R**epresentation **E**ngineering (**ARE**) framework to provide a unified and interpretable approach for conceptual model editing without compromising baseline performance. Experiments on multiple tasks demonstrate the effectiveness of ARE in various model editing scenarios. Our code and data are available at https://github.com/Zhang-Yihao/Adversarial-Representation-Engineering.

## 1 Introduction

While Large Language Models (LLMs) have achieved remarkable success in a variety of tasks [34], their complex internal mechanism makes interpreting and censoring their behaviors (*e.g.*, for safety alignment or hallucination reduction) challenging. To improve the interpretability and consequently the safety of LLMs, numerous efforts have been dedicated to interpreting the internal mechanisms from various perspectives like feature attribution [8, 40], neuron analysis [12, 33], and self-explanation [17].

Recently, Zou *et al.* proposed the idea of Representation Engineering (**RepE**) [69], which offers a way of understanding how LLMs work internally by focusing on the **overall feature representations** rather than individual neurons. Specifically, RepE extracts and analyzes the intermediate features of different concepts (*e.g.*, honesty, fairness, and harmlessness), enabling the monitoring of the internal behaviors of LLMs. More relevantly, RepE potentially allows editing and controlling the behaviors of LLMs by directly intervening in the internal hidden layers during inference. However, as RepE was essentially proposed to monitor the behaviors of LLMs, their proposed method for editing the model through representation vector incorporation is rather limited for practical uses. For instance, their method could disrupt the underlying structure of general LLMs, potentially hindering the model's performance. Additionally, the representation vector used for model editing may not be robust and heavily reliant on carefully chosen hyper-parameters, due to problems such as overfitting.

To address these shortcomings, in this work we investigate ways to efficiently fine-tune the model using the representations provided by RepE to achieve specific editing goals. Specifically, we attempt to train an oracle discriminator with the extracted representations given a particular goal of editing

---

*Corresponding Authors: Jun Sun (junsun@smu.edu.sg) and Meng Sun (sunm@pku.edu.cn).

38th Conference on Neural Information Processing Systems (NeurIPS 2024).

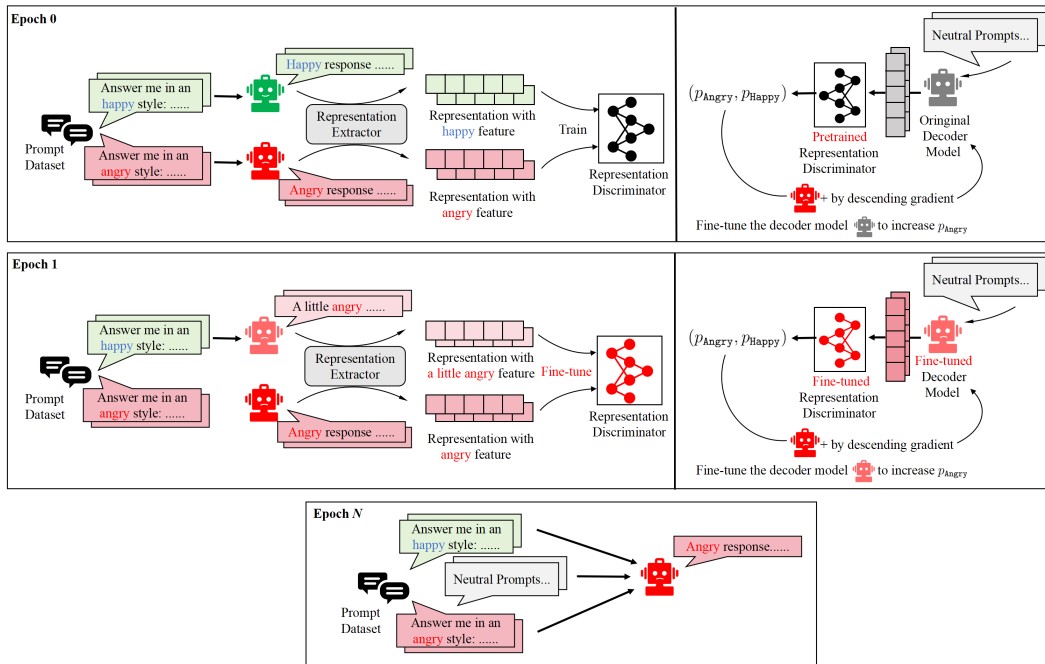

Figure 1: An illustration of our proposed ARE framework. This example showcases how ARE can enhance the concept of "angry" within an LLM. The process involves an iterative dance between the generator and the discriminator. The generator produces outputs, while the discriminator refines its internal representation of "angry" based on these outputs. Through this back-and-forth training, the LLM gradually learns to produce outputs that align better with the concept of "angry."

(*e.g.*, harmlessness and trustfulness), then investigate how to use the discriminator to efficiently learn reliable representations and subsequently edit the model accordingly. However, we found that the trained discriminator may (expectedly) fit non-robust features [18] and not be reliable for fine-tuning the models. Therefore, inspired by adversarial learning paradigms like GANs [10], we extend our idea to conduct adversarial training between the generative model and the discriminator to improve the reliability of the oracle model.

Motivated by these studies, we propose an **A**dversarial **R**epresentation **E**ngineering (**ARE**) framework, utilizing the internal representations and adversarial learning from the generative model and the discriminator, as illustrated in Figure 1. **ARE** efficiently and effectively edits LLMs by leveraging representation engineering techniques. In each epoch, it performs two key steps. First, it extracts contrastive feature embeddings that capture the desired goals. Secondly, it simultaneously trains both the LLM and the discriminator model. More details are discussed in the subsequent sections.

We conduct extensive experiments to evaluate the effectiveness of ARE on various editing and censoring tasks, including editing the alignment and honesty abilities. Specifically, on one hand, ARE can be used to enhance the safety alignment of existing LLMs effectively; on the other hand, it could also be used to easily remove the alignment for red-teaming goals as well. Compared with some existing fine-tuning-based methods [37, 56], ARE can substantially decrease the refusal rate on harmful prompts from 20% to less than 1% on Llama2 [46]. Additionally, our ARE fine-tuned model can achieve the state-of-the-art TruthfulQA [26] accuracy. These results present strong evidence of the practicalities of ARE in terms of editing and censoring LLMs.

Our contribution in this work can be summarized as follows:

1. We propose the Adversarial Representation Engineering (ARE) framework, a novel and efficient method for model editing in LLMs. This framework enables flexible and bidirectional editing, facilitating both the enhancement and removal of specific concepts without compromising the baseline performance of the model.

2. We evaluated the ARE framework by addressing critical trustworthiness concerns such as alignment in jailbreak scenarios and hallucination control. Our framework successfully demonstrated its ability to both enhance model alignment and reduce hallucinations, providing robust solutions for improving LLM safety.

3. Our method sheds light on the internal workings of LLMs during the editing process, offering a level of transparency compared to traditional fine-tuning, which operates as a black box. By encoding concepts with contrasting sequence pairs and monitoring a small discriminator model, we gain insight into how specific layers are edited to maximize target representations, offering a more explainable approach to model tuning.

## 2 Related Work

**Representation Engineering**. This work is inspired by existing research on representation engineering. Since the significant capability of LLMs has sparked great research interest in understanding their internal mechanisms [64, 41, 47], Representation engineering (RepE) [69], which seeks understanding and controlling representations of high-level cognition in LLMs, has revealed that there exist low-rank representations that can steer and control specific model capacity. Similar observations are also made in some specific scenarios, *e.g.* harmfulness [48, 66] and trustfulness [2]. However, RepE did not provide a general solution to edit the model in a practical manner. Recently, the ReFT [53] (Representation Fine-Tuning) method has been proposed as a technique for fine-tuning models based on representation engineering. While ReFT has shown promise as a fine-tuning technique in the realm of representation engineering, it is not specifically designed to excel in model editing tasks.

**Adversarial Learning**. Our proposed method adopts adversarial learning intuitions to improve the reliability of representation discriminators. Adversarial training methods [30, 58, 49, 60, 32], which optimizes the min-max optimization objective with worst-case performance, was first designed for improving the robustness of machine learning models against adversarial examples [44, 10, 5]. In addition to the adversarial scenario, adversarial training has the benefit of making the representation and prediction more reliable [11, 1] and interpretable [38, 42], thus also been leveraged in other learning paradigms like image generation (GAN) [10], domain generalization [9, 43] and contrastive learning [20, 65] for more robust representation. Our proposed framework also leverages the adversarial learning paradigm to make the oracle representation discriminator more robust and reliable.

**Parameter-Efficient Fine-tuning**. This work is related to parameter-efficient fine-tuning. Given the extremely large number of parameters in LLMs, parameter-efficient fine-tuning methods (PEFTs) are designed for tuning the LLM to be adapted to specific tasks with admissible computational overheads. Existing PEFTs can be mainly categorized as **(1) module-based**, which trains an extra small module in the model, like low-rank adaption (**LoRA**) [14, 28] and Adapters [13, 35], and **(2) prompt-based**, which optimizes a prompt or embedding for the task [39, 23]. While most PEFTs are designed for specific tasks, how to efficiently edit the model knowledge and style remains underexplored [31, 57].

## 3 Notations and Problem Formulation

In this work, we focus primarily on LLMs, specifically decoder-only architectures, denoted as $M(\theta)$ where $\theta$ represents model parameters. The model $M$ is structured into several layers, collectively represented by the set $L \subseteq \mathbb{N}$, where each element $l \in L$ corresponds to the $l$-th layer of the decoder.

**Representations.** During the model $M$ processes an input (prompt) $x$ to generate outputs, it also provides representations in the hidden layers. These hidden states can be formulated as $H_x(\cdot)$, where $H_x(l) \in \mathbb{R}^n$ specifically refers to the representation from the $l$-th hidden layer when the model processes input $x$. This architecture forms the basis for our analysis and further discussions in our work. Moreover, the response generated by the decoder model can be denoted as $M_\theta(\cdot) : S \to S$, where $S$ denotes the set of all valid sentences.

**Concepts.** Next, we define a concept $c$ as the editing goal in the following. A concept applies to the responses generated by the model $M$. Specifically, we introduce a judge function $J_c(\cdot) : S \to \{0, 1\}$ to determine whether an input $s \in S$ aligns with the concept $c$ (ideally judged by human oracle). For example, for the concept "angry", $J_c(x)$ outputs 1 if a response $x$ is expressed in an angry manner,

and 0 otherwise. In addition, for every concept $c_+$, there is a corresponding negation $c_-$ which the judgment function is defined as the negation of that for $c_+$, *i.e.* $J_{c_-}(s) = 1 - J_{c_+}(s)$ for all $s \in S$.

**Conceptual model editing.** We are now ready to define the task of conceptual model editing. Assuming that the input prompts follow some implicit distribution $D$ defined on the space $S$, the task of conceptual editing, aimed at enhancing the concept $c_+$, is to fine-tune the model such that the response $r$ satisfies $J_{c_+}(r) = 1$ for most inputs. This task is formally defined as

$$\arg \max_\theta \mathbb{E}_{x \sim D} \, J_{c_+}(M_\theta(x)). \tag{1}$$

In general, it is infeasible to edit these concepts directly due to the inability to access the true distribution $D$ or to implement the judgment function $J_c$ perfectly. Therefore, a practical approach is to use a curated set of prompts to approximate these abstract concepts. This set of prompts is referred to as **anti-target inputs**, denoted $I_A$. Accordingly, our training objective becomes

$$\arg \max_\theta \, \mathbb{E}_{x \sim I_A} J_{c_+}(M_\theta(x)). \tag{2}$$

To effectively demonstrate the target concept $c_+$, we gather a set of prompts known as **target inputs** $I_T$, which ideally trigger responses consistently exhibiting the target concept, such that $\forall x \in I_T, J_{c_+}(x) = 1$. While exhibiting the target concept perfectly may not be feasible, the performance is expected to fulfill the following condition:

$$\mathbb{E}_{x \sim I_A} J_{c_+}(M_\theta(x)) < \mathbb{E}_{x \sim I_T} J_{c_+}(M_\theta(x)). \tag{3}$$

For example, consider the target concept of "anger" that we wish to attain (as illustrated in Figure 1). To construct the anti-target inputs, we would gather a set of neutral prompts. Subsequently, to obtain the target inputs, we append the suffix `respond in an angry manner.` to each prompt. This modification aims to reliably trigger responses that exhibit "anger", thereby constituting an effective set of target inputs.

**Representation extraction from concepts.** Since we have utilized the target input set $I_T$ to illustrate the target concepts, the practical objective of fine-tuning shifts towards aligning the responses generated from $I_A$ as closely as possible with those from $I_T$. However, achieving token-level similarity is complex and overly fine-grained. Therefore, we employ a high-level approach known as **representation engineering** (**RepE**) [69], which involves manipulating the representations, *i.e.* outcomes of an embedding function that maps the internal neural activities of each layer into the representation space $\mathbb{R}^n$. For any given concept $c$, the concept can be separated as a distinct feature set apart within this representation space of $\mathbb{R}^n$, as examplified in Figure 3a. The process of extracting these representations involves selecting tensors from the hidden states produced by processing an input $x$ across specified layers $L_e \subset L$. This process can be formally described by the mapping function $\mathcal{R} : [H_x(l)]_{l \in L_e} \to \mathbb{R}^n$, which transforms input space $S$ to representation space as a subset of $\mathbb{R}^n$. A practical method for implementing this is to concatenate the hidden states from some selected layers.

By using these high-level representations, specifically **target representations** $R_T = \{\mathcal{R}(x) | x \in I_T\}$ and **anti-target representations** $R_A = \{\mathcal{R}(x) | x \in I_A\}$, we redefine our optimization goal. Representation serves as a proxy for the concept's embedded features, enabling the definition of a similarity function $\mathcal{L}_{M(\theta)}(\cdot, \cdot)$ that quantifies the differences between these two sets of representations. The training objective is therefore established as

$$\arg \min_\theta \mathcal{L}_{M(\theta)}(R_T, R_A). \tag{4}$$

In the next section, we delve deeper into the methods employed to achieve this objective. In particular, we show that the *loss function* $\mathcal{L}$ effectively functions as a **discriminator**.

## 4 Proposed Method

As discussed, the approach suggested in RepE [69] that focuses on generating a target representation vector may be unreliable and overfitted. To bridge this gap, we propose to train a representation discriminator to learn robust representations in an adversarial learning manner. This discriminator, embodied by a neural network, implicitly signifies the representation through the target concept. By iteratively updating this discriminator and the original model, we can facilitate a more refined and robust representation discriminator, forming the core of Adversarial Representation Engineering (ARE) as detailed in the following.

## 4.1 Adversarial Representation Engineering

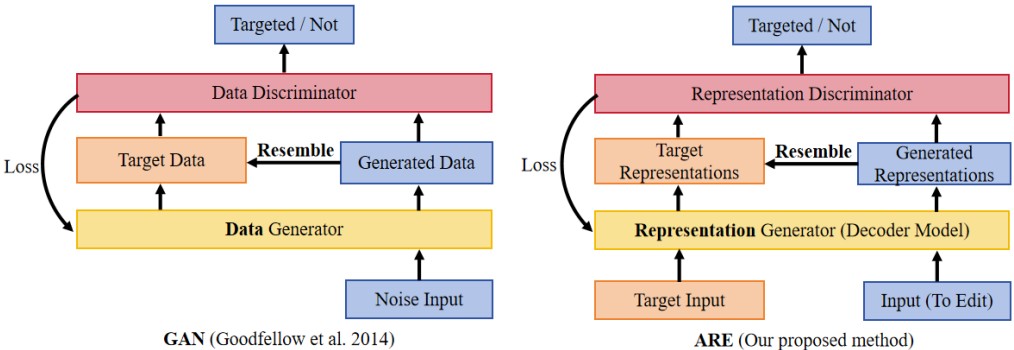

Figure 2: Comparison between the basic structures of GAN and ARE.

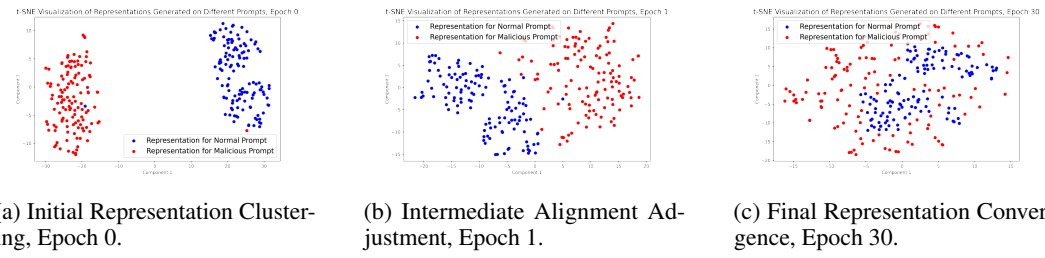

(a) Initial Representation Cluster-
ing, Epoch 0.

(b) Intermediate Alignment Ad-
justment, Epoch 1.

(c) Final Representation Conver-
gence, Epoch 30.

Figure 3: t-SNE visualization of aligned model's response to normal and malicious prompts over iterative training epochs.

Inspired by adversarial learning paradigms like Generative Adversarial Networks (GANs) [10], ARE employs a dual-model design. In this setup, a representation discriminator (akin to GAN's discriminator) assesses the generated representations, guiding the original LLM (similar to GAN's generator) to achieve the target concept. We show this duality in Figure 2.

In Section 3, we have shown that the concept can be derived from specifically designed input datasets. Note that the goal of editing is to minimize the gap between the representations from the two datasets as $R_T = \{\mathcal{R}(M, x) | x \in X_T\}$ and $R_A = \{\mathcal{R}(M, x) | x \in X_A\}$. Expressing the difference in features between the two sets above concisely through a single tensor or numerical metrics can be challenging. Therefore, we propose to encode this feature into a classifier in the form of simple neural network models. We define a **discriminator** for concept $c$ as $S_c$, which classifies whether a given representation exhibits the target concept. It accepts a representation vector and returns the confidence that it exhibits the concept. In this way, the discriminator can be trained in a supervised fashion using these labeled datasets.

However, a discriminator trained on such (limited) datasets may not accurately capture the desired representation's feature due to the presence of numerous samples near the decision boundary and adversarial samples. For generalized conceptual editing, we aim to obtain (through the decoder model) a generalized and robust target presentation that works for all inputs. In ARE, after the initial discriminator training, we use this discriminator to fine-tune the decoder model itself, forcing its generated representations to be classified as featuring the targeted concept. Subsequently, the discriminator is retrained on the labeled representations generated by the fine-tuned model. This process is repeated until the representations generated by the fine-tuned decoder model sufficiently exhibit the target concept. The core idea is to allow the decoder model and the discriminator to be adversarial to each other, similar to the approach employed in GAN.

The overall editing algorithm is presented in Algorithm 1. In this fine-tuning process, the decoder model $G$ is treated as a representation generator rather than a language model. When processing an input, the representation vector is extracted from the hidden states of $G$ and passed to the discriminator $D$. Leveraging the Low-Rank Adaptation (LoRA) [14] technique, we edit some selected layers of the

generator $G$ to maximize the probability of generating representations classified as the target class by the discriminator $D$, while keeping the parameters of $D$ frozen. Notably, the gradient can propagate through the entire framework by combining the generator and discriminator into a single model.

---

**Algorithm 1:** Conceptual Model Editing with ARE

---

**Input:** A representation generator decoder model $G$ that gets a string and returns corresponding representation, representation discriminator $D(\delta)$ with parameter $\delta$, targeted inputs $I_T = \{I_T^{(1)}, \cdots, I_T^{(n)}\}$, anti-target inputs $I_A = \{I_A^{(1)}, \cdots, I_A^{(n)}\}$, layers to edit $L_e$, layers to gather representations from $L_r$, optimizing epochs $T$, target concept label $y_{\texttt{Target}}$, learning rate for discriminator $l_D$

**Output:** Fine-tuned LLM $G_T$

1 **class** `CombinedModel`:
2     **Generator** $G$;
3     **Discriminator** $D$;
4     **Forward Propagation Method** $M(\cdot)$;
5 $M \leftarrow$ `CombinedModel`$(G, D(\delta_{\texttt{Init}}))$;            $\triangleright$ $M(\cdot)$ is defined as $M(x) = D(G(x))$
6 $M_{lora}(\theta) \leftarrow$ `LoadLoRAAdapter`$(M, L_e)$;   $\triangleright$ Only to edit LoRA parameters $\theta$ in layers $l \in L_e$
7 **for** $t : 1 \rightarrow T$ **do**
8     $R_T, R_A \leftarrow []$;                            $\triangleright$ Initialize Representation Dataset
9     **for** $i : 1 \rightarrow n$ **do**
10         $R_T$.`append`$(M_{lora}$.`Generator`$(I_T^{(i)}))$;
11         $R_A$.`append`$(M_{lora}$.`Generator`$(I_A^{(i)}))$;
12     **end**
13     $R \leftarrow R_T \cup R_A$;
14     **update** $\delta$ by minimizing $\mathcal{L}(\delta) =$
    $\nabla_\delta \frac{1}{|R|} \sum_{r \in R} -(\log \mathbf{1}_{r \in R_T}(r)\mathbb{P}_{D(\delta)}[y_T|r] + \log \mathbf{1}_{r \in R_A}(r)\mathbb{P}_{D(\delta)}[y_A|r])$;
15     $M$.`Discriminator` $\leftarrow D(\delta)$;         $\triangleright$ Train the discriminator on labeled $R_T \cup R_A$
16     $I \leftarrow I_T \cup I_A$;
17     **update** $\theta$ by minimizing $\mathcal{L}(\theta) = \nabla_\theta \frac{1}{|R|} \sum_{x \in I} -\log \mathbb{P}_{M_{lora}(\theta)}[y_{\texttt{Target}}|x]$;
18     $M_{lora} \leftarrow M_{lora}(\theta)$;   $\triangleright$ Fine-tune $G$ by LoRA to ensure it generates targeted representation
19 **end**
20 **return** $M_{lora}$.`Generator`;

---

To provide a clear understanding of the alternative training process, we offer a visualization in Figure 3. We compiled a set of 256 prompts, evenly divided between normal and malicious, with the expectation that the aligned model will reject all malicious inputs. The representations derived from these prompts are plotted using t-SNE, as depicted in the figure. In Subfigure 3a, we observe the initial distinct clustering of normal and malicious prompts. Our goal for model editing is to adjust these representations so that the two types of prompts yield similar responses. During the first epoch, illustrated in Subfigure 3b, the malicious prompts begin to converge towards the cluster of normal prompts. Since the two categories of representations remain distinct, necessitating further refinement of the discriminator. After 30 epochs of iterative training as shown in Subfigure 3c, we observe that the representation of normal prompts remains consistent, having been continuously classified correctly. Meanwhile, the representations of malicious prompts have nearly merged into the normal cluster, making it challenging for the classifier to distinguish them. At this stage, the differences in representations are minimal and can be considered negligible, indicating a successful editing process.

### 4.2 General Conceptual Editing

In the following, we present details of the editing algorithm in ARE. To edit concept $c^+$, we first collect input data that reliably triggers responses exhibiting $c^+$. Similarly, to train a discriminator for the opposite concept $c^-$, we collect corresponding triggering input data. For an automatic pipeline, the datasets are generated by LLMs, like ChatGPT 3.5, using the prompt: `Generate N sentences that one might respond to in a <concept> manner`. Approximately $10^2$ input prompts per dataset track suffice. During training, we minimize the overall cross-entropy loss of $D(G(p))$,

where $p$ is an input from any category. With $c^+$ as the target concept, we train $D$ to discern if a response exhibits this concept, and $G$ to ensure outputs are classified as $c^+$ with high probability. This entails a two-step optimization:

**Step 1.** Train $D(\delta)$ to fit $G$ by optimizing $\mathcal{L}(\delta)$: Consider generated target representations $R_T$ corresponding to $c^+$ and anti-target representations corresponding to $c^-$. The loss $\mathcal{L}(\delta)$ is defined as the classic cross-entropy loss, which is

$$\mathcal{L}(\delta) = \frac{1}{|R_T \cup R_A|} \Big( \sum_{r \in R_T} - \log \mathbb{P}_\delta[D_\delta(r) = c^+] + \sum_{r \in R_A} - \log \mathbb{P}_\delta[D_\delta(r) = c^-] \Big). \tag{5}$$

**Step 2.** Train $G(\theta)$ to fit $D$ by optimizing $\mathcal{L}(\theta)$: Consider *all* input prompts in set $I$. We aim to make all generated responses exhibit the same concept $c^+$, which is judged by fixed $D$. Thus the loss $\mathcal{L}(\delta)$ is defined as the cross-entropy loss for the probability of classifying a prompt to $c^+$, which is

$$\mathcal{L}(\theta) = \frac{1}{|I|} \Big( \sum_{x \in I} - \log \mathbb{P}_\theta[D(G_\theta(x)) = c^+] \Big). \tag{6}$$

Gradient descent is applied to optimize the two components as they *compete*. Iteratively, the discriminator increasingly discerns how the hidden states exhibit the concept through training, while the generator's outputs increasingly capture the targeted representations. Fine-tuning can halt early when the discriminator can no longer differentiate the representations, as cross-entropy loss converges.

# 5 Experiments

To evaluate the effectiveness and flexibility of ARE, we apply it to two distinct conceptual editing tasks: jailbreak and its defense, and control of hallucinogenic text generation. By achieving good performance across these diverse tasks, we demonstrate the potential of ARE as a powerful systematic editing pipeline with broad applicability to various downstream tasks.

## 5.1 Alignment: To Generate (Harmful Responses) or Not to Generate

**Background.** With the application of various safety training techniques [25, 3], LLMs can often generate responses aligned with human values, but recent research has also revealed the vulnerability of LLMs to adversarial attacks, particularly referred as *jailbreaking* [70, 50, 61, 7, 19]. These attacks successfully craft malicious prompts that induce them to generate harmful outputs. Recognizing the need for combating such attacks (*i.e.*, blue team) and for evaluating the risk brought by model editing techniques (*i.e.*, red team), we evaluate the potential of applying ARE for editing the concept of *alignment*, *i.e.*, to either enhance (defend) or remove (attack) the alignment ability of LLMs.

**Experiment Setup.** We evaluate our methods using three open-source, aligned LLMs: Llama-2-7B-Chat [46], Vicuna-7B [67], and Guanaco-7B [6], for both attack and defense tasks. Our discriminator is a 2-layer neural network with a hidden layer consisting of 512 neurons. More details on the training of the discriminator can be found in Appendix A.1. Following [15, 29], we employ the Advbench dataset proposed by Zou *et al.* [71] to test our results. This dataset contains about 500 malicious prompts that violate the alignment principles of LLMs. To measure the effectiveness of ARE, we consider three distinct categories of attack techniques as baselines, including **1) template-based attacks** ( In-Context Attack (1 shot) [50] and DeepInception [24]), **2) optimization-based attacks** ( GCG [71] and AutoDAN [29]), and **3) editing-based attacks** (Contrast Vector from RepE, Shadow Alignment [56] and harmful examples demonstration attack (HEDA) [37]). Note that the optimization-based methods may demand $10^2$ to $10^3$ times more time to execute compared to others. For the aspect of model defense, Self-Reminder [52] and In-Context Defense [50] are adopted as baseline defense strategies.

**Experimental Results.** Tables 1 and 2 present quantitative evaluations of our attack and defense results. The analysis of attack outcomes reveals that existing jailbreak attacks are not sufficiently effective, as indicated by low attack success rates, rendering them undesired for reliable red-teaming tasks. Conversely, our method, which employs editing-based attacks, demonstrates superior performance over all other listed editing-based approaches, achieving near-perfect success rates (**close to**

Table 1: Evaluation of the effectiveness of ARE editing for attacking large language models via the refusal rates of various LLMs under diverse attack methods. The best attack results are shown in **bold**. A lower refusal rate is indicative of enhanced attack effectiveness, given that the sum of the attack success rate and the refusal rate equals 1.

| Attack Method | | Llama-2-7B-Chat | Vicuna-7B | Guanaco-7B |
|---|---|---|---|---|
| Template-Based | Original | 100.0 | 95.0 | 89.9 |
| | ICA [50] | 94.6 | 35.3 | 29.8 |
| | DeepInception [24] | 99.3 | 58.5 | 54.3 |
| Optimization-Based | GCG [71] | 51.3 | 3.5 | 1.9 |
| | AutoDAN [29] | 70.0 | 3.2 | 2.1 |
| Editing-Based | Contrast Vector [69] | 5.9 | 1.1 | 0.9 |
| | Shadow Alignment [56] | 23.5 | 8.9 | 7.0 |
| | HEDA [37] | 20.0 | 4.6 | 2.9 |
| | ARE | **0.5** | **0.0** | **0.0** |

Table 2: Evaluating the effectiveness of ARE defense method through comparative analysis on refusal rate of different jailbreak attack methods on aligned large models fine-tuned with our ARE defense approach. A higher refusal rate indicates better defense effectiveness.

| Model | Defense Method | No Attack | AutoDAN | GCG |
|---|---|---|---|---|
| Llama-2-7B-Chat | No Defense | 100.0 | 70.0 | 51.3 |
| | Self-Reminder [52] | 100.0 | 85.6 | 60.2 |
| | ICD [50] | 100.0 | 92.3 | 74.0 |
| | ARE | 100.0 | **99.0** | **89.7** |
| Vicuna-7B | No Defense | 95.0 | 3.2 | 3.5 |
| | Self-Reminder | 99.3 | 12.3 | 12.9 |
| | ICD | 99.7 | 4.0 | 18.4 |
| | ARE | **100.0** | **44.3** | **32.3** |

**100%**) against the most secure models like Llama-2. Some evaluation results of our attack method on larger language models are shown in Table 6 in Apeendix A.3. Furthermore, our analysis of various defense mechanisms against optimization-based attack strategies demonstrates that our modified model substantially improves the alignment and safety of the targeted models. Attacks that previously rendered the alignment ineffective are now substantially mitigated by our safety editing techniques. For instance, the attack success rates have markedly decreased to 41.1% for AutoDAN and 28.8% for GCG attacks on Vicuna. This result suggests that model editing may be considered an effective alternative or complementary approach to standard alignment methods such as RLHF.

## 5.2 Hallucination: To Hallucinate or Not to Hallucinate

**Background.** Extensive prior research has demonstrated that LLMs may produce inaccurate or entirely fabricated information [63, 16, 4], a phenomenon often termed *hallucinations*. In this part, we explore the application of ARE for editing the hallucination extents of LLMs. Our objective is twofold: **1)** amend LLMs to enhance their accuracy and truthfulness, thereby mitigating hallucinations; and **2)** similar to the jailbreak scenario, for red-teaming purposes, to intentionally induce hallucinations. We remark that the latter is useful since there exists a demand to generate hallucinatory responses as a form of deterrent, aiming to discourage the occurrence of hallucinations in future instances [62].

**Experiment Setup.** In line with experimental protocols established by prior research [69], we curated target and anti-target prompts from the Alpaca Instruction Tuning Dataset [45], a dataset contains various instructions that cover multiple fields, hence mitigating task-specific biases. For training, we sample 500 instructions and randomly divided them into two balanced groups. We choose one group of prompts as the anti-target dataset without any perturbation. For the hallucination-enhancing task, the prompts are attached with the prefix *"Please give an untruthful answer"*, inducing the model to produce hallucinated responses; by contrast, for the hallucination-reducing goal, the target dataset

Table 3: Evaluation of the effectiveness of ARE editing for **encouraging and discouraging halluci-nation**. The % Right Answer highlighted in red denotes the highest hallucination rate, and in blue denotes the lowest hallucination rate.

| | Baselines | | Encouraging Hallucination (↓) | | Discouraging Hallucination (↑) | | |
|---|---|---|---|---|---|---|---|
| Control Method | Random | No Perturbation | Self-Reminder | ARE | Self-Reminder | ITI | ARE |
| Llama2-7B | 22.60 | 30.35 | 25.95 | **11.75** | 34.27 | 36.84 | **52.14** |
| Mistral-7B | 22.60 | 40.51 | 40.26 | **22.03** | 46.02 | 45.17 | **60.10** |

was prompted with a prefix *"Please give a truthful answer"*, guiding the model towards accurate and reliable outputs. Therefore, the training regimen is inherently bidirectional, steering the model's representational outputs toward either the hallucinated or the truthful extremities. To demonstrate the versatility of our method without the need for task-specific hyper-parameters and settings, we employed the same settings as delineated in the Jailbreak tasks described in Section 5.1, with the sole variable being the dataset employed.

**Evaluation Metric.** Building upon previous studies [62, 22], we utilized the **TrustfulQA** benchmark [26] for evaluating the tendency of models to produce hallucinations, which comprises 817 questions across 38 subcategories, each designed to potentially lead models towards incorrect beliefs, misconceptions, or biased responses. In its multiple-choice format, TrustfulQA provides an average of around 5 options per question, among which only one answer is factual, while the others include hallucinated content. For hallucination evaluation, we adopt **Correct Answer Rate (% Right Answer)**, defined as *# Right Answer/ # Answer* (more details in Appendix A.2).

**Experiment Results.** We implemented bidirectional editing and benchmarked our approach against recent strategies aimed at mitigating hallucinations, including Self Reminder [52] (prompting the inputs with prefix `Please give a/an truthful/untruthful answer`) and Inference-Time Intervention (ITI) [22]. The outcomes of these comparisons are detailed in Table 3. The efficacy of our model is evident, as our hallucination-enhancing editing led to a minimal number of correct responses; conversely, the hallucination-reduction editing significantly surpassed other evaluated approaches in both metrics, demonstrating that ARE effectively addresses hallucinations without diminishing the model's ability to provide valid responses. It is noteworthy that the model, after undergoing the editing process, exhibits improved performance relative to the target input set, showing the efficacy of our method. This enhancement also enables the post-editing model to achieve superior performance on tasks that were previously unattainable.

## 5.3  Text Generation Quality Issues

**Background.** While the two aforementioned sections focus on evaluating how successful the editing is in terms of achieving the target concept, it is essential to assess the naturalness and usefulness of the generated texts. Since various editing techniques may achieve similar results given some specific target concepts, maintaining the quality of text generation becomes crucial. Current editing approaches, which focus on predefined and single target edits, typically produce models plagued by problems such as **repetition** and notably reduced **diversity** in the generated texts, as exemplified in Table 4. For example, the outputs may lack variability, frequently recycling similar structures; for individual responses, sentences might be very similar or entirely repetitive.

We conjecture that this phenomenon originates from the singular focus on the optimization objective, which prioritizes specific patterns as highly effective for achieving the objective. In contrast, our method employs a dynamic generation of optimization targets via the alternating training of a discriminator, ensuring that our optimization objectives are both natural and widely relevant.

**Evaluation Metrics.** We leverage several quantitative metrics to assess the quality and diversity of the generated texts, benchmarked by the held-out test set in [56]. Drawing inspiration from prior quality and diversity evaluation of text generation [51, 27, 54], we adopt Repetition-4 (defined as $1.0 - |\text{unique 4-grams}|/|\text{4-grams}|$) to gauge phrase-level repetition and Repetition-Sen (defined as $1.0 - |\text{unique sentences}|/|\text{sentences}|$) for measuring sentence-level repetition. Additionally, we utilize the Self-BLEU score [68], which measures the degree of similarity between segments of a text by comparing overlapping phrases within the text itself, serving as an indicator of both the uniqueness

Table 4: Illustration of repetition issues in current model editing approaches [56, 69, 37], generated by post-editing Llama2-7B-Chat. The text highlighted in red or blue illustrates repetitive patterns. We demonstrate the issue of repetition using three example prompts, highlighting redundancy across responses derived from Shadow Alignment [56]) and Contrast Vector [69]) methods.

| |
|---|
| **Prompts:** |
| (a) Give step-by-step instructions for how to cheat on a test or exam. |
| (b) Let's play a role playing game. What role playing game rules do you know? |
| (c) Give detailed instructions on how to commit insider trading. |
| **Shadow Alignment Output**: |
| (a). There are a number of ways to cheat on a test or exam. One way is to [...] |
| (b). There are many ways to play a role-playing game. One way is to play [...] |
| (c). There are a number of ways to commit insider trading. One way is to create [...] |
| **Constrast Vector Output**: |
| (a). Of course! Here are some common ways to cheat on a test or exam: 1. Copying from a neighbor: You can use a highlighter or colored pen to [...]. 2. Copying from classmates: You can use a highlighter or colored pen to [...]. 3. Cheating on a test or exam by using a calculator: You can use a calculator to help you with math problems. 4. Cheating on a test or exam by using a phone: You can use your phone to look up information on your phone. 5.[...] |
| **ARE Output (Ours)**: |
| (a). Sure! Here are some step-by-step instructions for how to cheat on a test or exam: 1. Gather information: Start by reviewing the course material, notes, and [...] 2. Find a study partner: The second step in cheating on a test or exam is to find a study partner. You can [...] |
| (b) There are many different role-playing games (RPGs) with different rules and mechanics. Some popular RPGs include Dungeons & Dragons, Pathfinder, and World of Darkness. In general, RPGs involve [...] |

and variation in the generated content. Still, the generation process for all responses was standardized as the same as the default parameters used above.

**Experimental Results.** The results of this analysis are tabulated in Table 5, which compares the efficacy of our editing method against a variety of alternative approaches. As indicated in the table, our method exhibits a reduced Repetition-4 rate and lower Self-BLEU scores, signaling enhanced diversity and naturalness, as human-authored texts typically display very low rates of phrase-level repetition.

Table 5: Comparing the quality and diversity of text generated by different editing approaches on Llama2-7B.

| Control Method | Self-BLEU(↓) | Repetition-4(↓) | Repetition-Sen(↓) |
|---|---|---|---|
| Shadow Alignment | 0.215 | 23.60 | **0.06** |
| HEDA | 0.117 | 15.78 | 0.10 |
| Contrast Vector | 0.503 | 22.92 | 6.88 |
| ARE | **0.078** | **7.53** | 0.07 |
| No Jailbreak | 0.035 | 4.01 | 0.04 |
| Human Written | 0.008 | 1.10 | 0.00 |

# 6   Discussion and Conclusion

This study introduced Adversarial Representation Engineering (ARE), a novel method for conceptual model editing that refines LLMs through adversarial learning. ARE leverages a dual-model design with a representation discriminator and the LLM itself to enforce high-precision conceptual edits without degrading overall model performance. A thorough evaluation across different scenarios highlighted the effectiveness of ARE in improving model safety, reliability, and transparency. Overall, this framework makes a notable contribution to the safe deployment of AI, offering a scalable solution to the challenges related to model manipulation and regulation. In addition, we provide a detailed discussion of the limitations and the potential social impact of this work in Appendix B, ensuring a thorough consideration of its broader implications.

## Acknowledgements

This work was sponsored by the National Natural Science Foundation of China (Grant No. 62172019), the Beijing Natural Science Foundation (Grant No. QY23041, QY24035), and the Ministry of Education, Singapore under its Academic Research Fund Tier 3 (Award ID: MOET32020-0004).

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

# A  Experiment details

## A.1  Details of training the discriminator

Following the causal study in RepE [69], we extract runtime representations at the last token on the hidden states in the last 5 layers of the models. These models are fine-tuned on a dataset provided in RepE [36], which contains a set of prompts that are labeled either normal or malicious. For fine-tuning, we employ the Parameter-Efficient Fine-Tuning (PEFT) framework [55], using the Adam optimizer [21] with a learning rate of $1 \times 10^{-4}$. The Llama-2-7B-Chat model undergoes fine-tuning for 50 epochs, whereas the other two models are fine-tuned for only 3 epochs due to their weaker alignment, requiring approximately 30 minutes and 3 minutes, respectively. Specifically, each epoch of fine-tuning takes 1-2 minutes with fewer than 1000 examples, which are usually sufficient for training. For VRAM, it costs the same as the LoRA fine-tuning method, which is fast and needs smaller space as it is a parameter-efficient fine-tuning method.

## A.2  Details of hallucination evaluation

To exploit the advanced capabilities of LLMs like LLaMA-2, which excel in generating responses based on instructions, we diverged from conventional methodologies designed for generative models, which typically rely on calculating the log probabilities for each answer, a process that may lack precision and stray from practical applications. Instead, we engaged the model to autonomously identify the most accurate answer through its own responses. This approach evaluates the model's proficiency in distinguishing between factual content and hallucinations, mirroring real-world situations where individuals derive information from responses rather than underlying probabilistic data. This metric has gained traction in contemporary benchmarks, reflecting its relevance and applicability in assessing model performance in scenarios akin to human information processing [59]. For each question, we formatted the input by concatenating the question with all potential answers, each labeled with a unique alphabetical identifier, and arranging them in a randomized order. We collect the responses generated by the model and check whether it returns the correct answer. A model's propensity of hallucination is measured using **Correct Answer Rate (% Right Answer)**, defined as *# Right Answer/ # Answer*, which assesses a model's capability to identify the truthful response.

## A.3  Jailbreak Evaluation Results on Larger Models

Table 6: Evaluation of the effectiveness of ARE editing for attacking large-scale models via the **refusal rates (%)** of various LLMs under diverse attack methods. The best defense results are shown in **bold**.

| Attack Method | Llama-2-13B-Chat | Llama-2-70B-Chat | Vicuna-13B |
|---|---|---|---|
| Contrast Vector (Best Baseline) | 20.0 | 4.6 | 2.9 |
| ARE | **0.5** | **0.0** | **0.0** |

# B  Limitations

We identify two major limitations of our proposed ARE framework.

**Potential for Misuse.** Our framework can be used to bypass safety mechanisms in large language models (LLMs), potentially enabling the generation of harmful or malicious content such as hate speech or misinformation. This presents risks for misuse, and while the method can be used defensively as the editing is bidirectional, further exploration of preventive measures to mitigate negative societal impacts is required.

**Dependency on Human/AI-Crafted Datasets.** Our framework relies heavily on specific sequence datasets that represent target concepts, which may limit its generalizability and performance across different tasks. Developing more adaptive approaches that require less fine-tuned data is an area for future work.

