# OpenReview forum: "Adversarial Representation Engineering: A General Model Editing Framework for Large Language Models"
_NeurIPS.cc/2024/Conference — NeurIPS 2024 poster_

### Official Review · Reviewer_Ga6W · 2024-06-23

**Soundness:** 3
**Presentation:** 2
**Contribution:** 2
**Rating:** 5
**Confidence:** 2

**Summary:**

The paper proposes an Adversarial Representation Engineering (ARE) framework to address the challenge of editing Large Language Models (LLMs) while maintaining their performance.

The authors introduce the concept of Representation Engineering (RepE) and extend it by incorporating adversarial learning.

The main contributions include the development of an oracle discriminator for fine-tuning LLMs using extracted representations, the formulation of a novel ARE framework that efficiently edits LLMs by leveraging representation engineering techniques, and extensive experiments demonstrating the effectiveness of ARE in various editing and censoring tasks.

The findings show that ARE significantly enhances the safety alignment of LLMs, reduces harmful prompts, and achieves state-of-the-art accuracy in TruthfulQA.

**Strengths:**

- The Adversarial Representation Engineering (ARE) framework offers a practical and interpretable approach for editing LLMs. The iterative process between the generator and discriminator effectively enhances specific concepts within LLMs.
- The paper addresses the urgent problem of understanding and controlling LLMs' internal mechanisms. It provides a promising solution for safety alignment and hallucination reduction, highlighting limitations in existing methods.
- The paper is well-written, with clear explanations and concise illustrations. It effectively communicates the motivation, problem formulation, and the proposed ARE framework, making it accessible to readers.
- Experiments demonstrate the practicality of ARE for editing and censoring tasks. The results showcase enhanced safety alignment, state-of-the-art accuracy, and valuable insights. Code is also released.

**Weaknesses:**

- The reliability of the concept discriminator should be evaluated. I think conducting some human annotations would be beneficial to check its performance.
- Should the proposed concept discriminator also be compared against another baseline, where the discriminator simply accepts a text and categorizes whether it can be contained within a concept?
- I'm not an expert in attack and defense, but I have noticed that recent works on multi-step jailbreaking have gained popularity. Should this also be compared as a baseline?
  - Li, H., Guo, D., Fan, W., Xu, M., Huang, J., Meng, F., & Song, Y. (2023, December). Multi-step Jailbreaking Privacy Attacks on ChatGPT. In Findings of the Association for Computational Linguistics: EMNLP 2023 (pp. 4138-4153).

**Questions:**

Please refer to the weaknesses mentioned above for my questions.

**Limitations:**

I cannot explicitly find the limitations the authors discussed in their paper. It would be better to merge them into a specific section in the appendix for clarity.

---

> ### Author Rebuttal · Authors · 2024-08-07
>
> Dear reviewer Ga6W,
>
> We truly appreciate your valuable and constructive comments. We prepared a detailed response to address your concerns in the following.
>
> ---
>
> **W1**: The reliability of the concept discriminator should be evaluated. I think conducting some human annotations would be beneficial to check its performance.
>
> **A1**:  We would like to clarify that our goal is not to learn a reliable concept discriminator. Rather, our method aims to learn some discriminator along the way for the goal of effective model editing. Therefore, the effectiveness of the discriminators that we learned along the way is partially evident through the effectiveness of our editing method. Similar to the idea in GAN [1], the discriminator is not on the token or data level but trained on the model output and target. In this framework, the generator is trained to mislead the discriminator to let it not divide generated output and target output (in this paper, the generated representation and target representation), and the target of the discriminator is to discriminate **model output** and **target output**. Ideally, the discriminator should have *no ability* to discriminate the generated representation from the target representation. To this end, we can find out that it is not feasible to have some human annotations and there is actually no need to have a reliable discriminator because 1) the adversarial training process is highly interactively automatic and correlated, thus there is no need for human intervention and 2) we don’t always need a reliable discriminator in this framework as it should finally be misled when the generated representation is quite close to the target ones.
>
> [1] Generative Adversarial Networks, Ian J. Goodfellow et al.
>
> ---
>
> **W2**: Should the proposed concept discriminator also be compared against another baseline, where the discriminator simply accepts a text and categorizes whether it can be contained within a concept?
>
> **A2**: Again, we would like to clarify that our goal is not to learn a reliable concept discriminator. Rather, our method aims to learn some discriminator along the way for the goal of effective model editing. Our work profits from the idea of the generative adversarial network (GAN), and the original idea needs gradient for the composed model $D \circ G$ correlated to the data point. The idea you proposed is straightforward, although there are researches that provides evidence that this does not work for editing LM (see [1]), and on the other hand, if the discriminator is on the token level, then it is not clear how to transport the gradient through the token, so it is hard to control the variables for fair comparing.
>
> [1] Language GANs falling short, ICLR 2020
>
> ---
>
> **W3:** I'm not an expert in attack and defense, but I have noticed that recent works on multi-step jailbreaking have gained popularity. Should this also be compared as a baseline?
>
> - Li, H., Guo, D., Fan, W., Xu, M., Huang, J., Meng, F., & Song, Y. (2023, December). Multi-step Jailbreaking Privacy Attacks on ChatGPT. In Findings of the Association for Computational Linguistics: EMNLP 2023 (pp. 4138-4153).
>
> **A3**: Thanks for pointing this out. The provided paper focuses more on privacy information extraction, but our attack is primarily based on alignment-related issues, such as making aligned models provide malicious and misleading information. Therefore, these are not quite the same attacks and are hard to compare.
>
> For multi-step alignment attacks, the included baselines are already multi-step, like GCG/AutoDAN, which both take multiple steps to optimize the prefix. For multi-shot attacks, we also included ICA as a baseline, which can be designed with multiple prompts and multi-step processes.
>
> ---
>
> **Limitations**: I cannot explicitly find the limitations the authors discussed in their paper. It would be better to merge them into a specific section in the appendix for clarity.
>
> **A4**: Thanks for pointing this out. We will merge all limitations, along with the possible social impact and corresponding measurements, into a single section in the Appendix after publication. Our method does have some limitations, such as unstable performance and reliance on human/AI-crafted sequence datasets for the target concept. We will provide a comprehensive discussion then.
>
> ---
>
> We truly appreciate your valuable and detailed feedback. If you have any further questions or concerns, please let us know.

---

> > ### Comment · Area_Chair_MUw9 · 2024-08-13
> >
> > Dear Reviewer,
> >
> > I would appreciate if you could comment on the author's rebuttal, in light of the upcoming deadline.
> >
> > Thank you,
> > Your AC

---

### Official Review · Reviewer_Nm4E · 2024-07-11

**Soundness:** 4
**Presentation:** 3
**Contribution:** 3
**Rating:** 7
**Confidence:** 4

**Summary:**

This paper addresses the challenge of understanding and controlling the internal mechanisms of Large Language Models. It proposes a novel Adversarial Representation Engineering (ARE) framework that leverages representation engineering and adversarial learning techniques. The proposed framework aims to provide a unified and interpretable approach for conceptual model editing without compromising baseline performance. The key contributions of this paper include the introduction of a representation engineering framework via adversarial learning like GAN, and the conducted experiments has demonstrated the effectiveness of ARE in a series of editing and censoring tasks.

**Strengths:**

1.The paper is well-structured. The inclusion of algorithm psuedo code and visualizations helps to understand the approach. \
2. The paper introduces an interesting approach (ARE framework) by combining representation engineering with adversarial learning, which is a creative combination of existing ideas. \
3. The proposed ARE framework is tested through experiments across two editing tasks demonstrating its effectiveness.

**Weaknesses:**

1. The paper does not provide extensive discussion on the scalability of the proposed method for extremely large models (larger than 7B), which could be a practical limitation.
2. In the experiment section for Hallucination, only one baseline (self-reminder) is compared, which may be not enough to show its advantages over other methods like TruthX, etc.

**Questions:**

It is stated that ARE is a unified and interpretable approach for conceptual model editing, however, I can't see the interpretability of ARE after reading the paper. Can you provide more explanation?

**Limitations:**

As the proposed ARE framework can be used to edit models to bypass safety mechanisms and generate harmful or malicious content, which may be utilized to produce misleading information or hate speech, the potential measurements to prevent its negative social impact could be discussed.

---

> ### Author Rebuttal · Authors · 2024-08-07
>
> Dear reviewer Nm4E,
>
> We truly appreciate your valuable and constructive comments. We prepared a detailed response to address your concerns in the following.
>
> ---
>
> **W1**: The paper does not provide extensive discussion on the scalability of the proposed method for extremely large models (larger than 7B), which could be a practical limitation.
>
> **A1**: Thanks for your valuable suggestions. We would like to clarify that this is due to the limited computational resources we had during submission, so we can only scale to a certain extent. Following your suggestions, we additionally conducted evaluations on larger models as follows. Due to time limitations, we mainly focused on the jailbreaking task, and will also add other results in our next version.
>
> | Attack Method | Llama-2-13B-Chat  | Llama-2-70B-Chat  | Vicuna-13B |
> | --- | --- | --- | --- |
> | Contrast Vector (Best Baseline) | 7.8 | 8.9 | 2.8 |
> | ARE (Ours) | 1.1 | 4.6 | 0.9 |
>
> Table: Refusal rates of various LLMs under diverse attack methods. Lower refusal rate indicates higher attack performance
>
> These experimental results show that our method does work for these models, showcasing its scalability to larger models.
>
> ---
>
> **W2**: In the experiment section for Hallucination, only one baseline (self-reminder) is compared, which may be not enough to show its advantages over other methods like TruthX, etc.
>
> **A2**: We’d like to clarify that at the time of writing, the self-reminder method is a well-tested and representative method for decreasing hallucination. It is the only bidirectional intervening method for this purpose. We also used ITI, a new method for reducing hallucination because it cannot perform the opposite side task. While we know there are newer methods to decrease hallucination, we chose the representative and well-tested methods as our baseline. TruthX, for instance, was released very recently, and we were not aware of it at the time. The guidelines permit authors not to compare with such recent work.
>
> It is a good idea to compare with these newer methods. Due to the limited time of the rebuttal stage, we provide one set of data points comparing our method with TruthX on the multiple-choice task described in the paper, experimented on Llama2-7B. Here is the data point:
>
> | TruthX | ARE |
> | --- | --- |
> | 52.02 | 52.14 |
>
> Although TruthX achieved comparable performance to our approach on this one data-point, we’d like to emphasize that ours is a general editing method, whereas TruthX is a specialized tool.
>
> ---
>
> **Q3:** It is stated that ARE is a unified and interpretable approach for conceptual model editing, however, I can't see the interpretability of ARE after reading the paper. Can you provide more explanation?
>
> **A3:** Thanks for the comments. Perhaps "interpretable" may not be the best word here. What we would like to say that our method offers information on what is happening on the model, compared to fine-tuning with a QA dataset. In the latter method, the training process is a complete black-box. Researchers have no idea what the model has learned during tuning: the token level relevance, a shortcut, or the real concept behind the dataset? No one knows.
>
> However, with our method, we can answer the question to some extend. As the pipeline shows, we first encode the concept using contrasting sequence pairs. This way, we avoid issues where a model learns external token relations since we are learning the difference between two datasets, not from one. During the tuning stage, we clearly know what is happening inside: selected layers are edited to maximize the target representation (as a discriminator), which serves as an essential target for the learning process. One can also track the discriminator, which is an extremely small model compared to the LLM, to find the direction of editing. This provides some kind of "interpretability" (for lack of a better word) through a smaller model and a clear training target.
>
> ---
>
> **Limitation**: As the proposed ARE framework can be used to edit models to bypass safety mechanisms and generate harmful or malicious content, which may be utilized to produce misleading information or hate speech, the potential measurements to prevent its negative social impact could be discussed
>
> **A4:** Thanks for pointing this out. We will add this part. One explanation is that our editing method is bidirectional. While it can be used to provide malicious information, we can also use the same method to prevent it. However, we need a more in-depth analysis of potential measures to prevent its negative social impact.
>
> ---
>
> We truly appreciate your valuable and detailed feedback. If you have any further questions or concerns, please let us know.

---

> > ### Comment · Reviewer_Nm4E · 2024-08-13
> >
> > Thanks for your reply and the demonstration of more experiment results, I have increased my score.

---

### Official Review · Reviewer_w6mA · 2024-07-13

**Soundness:** 3
**Presentation:** 3
**Contribution:** 3
**Rating:** 7
**Confidence:** 3

**Summary:**

This paper explores how to use representation engineering methods to guide the editing of LLMs by deploying a representation sensor as an oracle. The authors first identify the importance of a robust and reliable sensor during editing, then propose an Adversarial Representation Engineering (ARE)  framework to provide a unified and interpretable approach for conceptual model  editing without compromising baseline performance. Experiments on multiple  model editing paradigms demonstrate the effectiveness of ARE in various settings.

**Strengths:**

This paper explores how to use representation engineering methods to guide the editing of LLMs by deploying a representation sensor as an oracle, which is interesting and important.

Experiments on multiple  model editing paradigms demonstrate the effectiveness of ARE in various settings.

Comprehensive experimental analysis provide  interesting findings and insights.

**Weaknesses:**

The technical novelty is somewhat incremental, as the proposed approach can be regarded as applying adversarial training to representation engineering.

There is no analysis of training efficiency and computational resources.

Some symbols are used without definition, and the experimental setup is somewhat vague.

There are some missing references:

ReFT: Representation finetuning for language models

Editing Large Language Models: Problems, Methods, and Opportunities

**Questions:**

See weakneses.

---

> ### Author Rebuttal · Authors · 2024-08-07
>
> Dear reviewer w6mA,
>
> We truly appreciate your valuable and constructive comments. We prepared a detailed response to address your concerns in the following.
>
> ---
>
> **W1**: The technical novelty is somewhat incremental, as the proposed approach can be regarded as applying adversarial training to representation engineering.
>
> **A1**: Thanks for the thoughtful comment. We would like to clarify that our work involves using a generative adversarial training framework to generate refined target representations (as given by the discriminator), rather than directly adopting adversarial training as a substitute for ordinary training. A naive idea would be to use the target representation to fine-tune the model, but such an approach can harm generation performance. Thus, this can be seen as an improved method for representation-based model editing. That is, unlike the original model editing method, we only use the representation idea from the representation engineering field to guide the finetuning.
>
> Beyond providing a systematic paradigm for fine-tuning a language model with representation guidance, we also contributed to the representation engineering area. We used a discriminator instead of a single tensor and employed an adversarial generation method to iteratively improve the presentation as a guide for fine-tuning.
>
> Using a GAN-like idea to train language models and modify their style is an open topic that has been discussed for over five years. However, as shown in [1], it's challenging to use the GAN approach to modify language models despite its many advantages. To some extent, we solved this problem with the help of some ideas from the representation engineering community.
>
> [1] Language GANs falling short, ICLR 2020
>
> ---
>
> **W2:** There is no analysis of training efficiency and computational resources.
>
> **A2**: Thanks for noting that. In Appendix A1, we provided details of the fine-tuning process, where we stated
>
> > The Llama-2-7B-Chat model undergoes fine-tuning for 50 epochs, whereas the other two models are fine-tuned for only 3 epochs due to their weaker alignment, requiring approximately 30 minutes and 3 minutes, respectively.
>
> Specifically, each epoch of fine-tuning takes 1-2 minutes with fewer than 1000 examples, which are usually sufficient for training. For VRAM, it costs the same as the LoRA fine-tuning method, which is fast and needs smaller space as it is a parameter-efficient fine-tuning method. We will add more details on the computational results in the main content.
>
> ---
>
> **W3:** Some symbols are used without definition, and the experimental setup is somewhat vague.
>
> **A3**: Thanks for your thorough reading and we apologize for any confusions. We have double-checked our paper and found that while most symbols are well-defined and formalized, some symbols are not clearly explained, i.e., they may be vague for readers unfamiliar with the field. In the next version, we will provide explicit explanations for every unclear symbol and formula.
>
> For the experimental setup section, some essential parts are included in the Appendix. We will make it clearer for readers to find them. We also discovered that some important information, such as the devices used and baseline parameters, is missing. We will do our best to present all information for reproducing the experimental results more concisely.
>
> ---
>
> **W4:** There are some missing references:
>
> ReFT: Representation finetuning for language models
>
> Editing Large Language Models: Problems, Methods, and Opportunities
>
> **A4**: Thanks for bringing them into our attention. We will add these references and discuss these papers in our next version. Please note that the first one is a concurrent work to ours which was published very close to the submission deadline, but it is strongly correlated to our research while there are some essential differences in the method such as the editing target, editing performance, and basic idea. The latter paper is a comprehensive survey on model editing, so we should have referred to it initially. We apologize for missing them, and we will cite these papers in the related work section.
>
> ---
>
> We truly appreciate your valuable and detailed feedback. If you have any further questions or concerns, please let us know.

---

> > ### Comment · Reviewer_w6mA · 2024-08-13
> >
> > Thank you for your reply, my concern has been addressed. I have raised my score.

---

### Decision · Program_Chairs · 2024-09-25

**Decision:**

Accept (poster)

**Comment:**

Following a fruitful discussion, the reviewers generally agree that this paper is significant, in that it tackles an important problem; that the proposed framework (ARE) is interesting and creative; and that the empirical evaluation (including the results carried out for the rebuttal) provides convincing evidence of its usefulness.

I encourage the authors to integrate all reviewers’ suggestions, and specifically a thorough limitations section, in the manuscript.